**communications** engineering

# Photonic fully-connected hybrid beamforming using microring weight banks

Mitchell Nichols [1] ✉, Hugh Morison [2], Armaghan Eshaghi[3], Bhavin Shastri [2] & Lutz Lampe [1]

Wireless communication at higher frequency bands has attracted research interest for fifth generation and beyond (5GB) wireless networks due to the large amount of unused bandwidth at these frequencies. However, there are substantial challenges associated with higher frequency bands due to the high path loss of the propagation environment and the high power consumption of the transceivers. Hybrid beamforming with massive multiple-input multiple-output (MIMO) has emerged as a solution to these problems by combining the performance and flexibility of digital beamforming with the energy efficiency of analog beamforming. Optical beamforming has recently been considered as an alternative to implement the analog component of a hybrid beamformer, which may offer improvements in size, weight and power consumption in comparison to conventional electronics. This paper proposes a new approach to implement an optical beamforming system based on photonic vector modulators using tunable photonic filters. Our experimental demonstration of the proposed optical beamformer shows that microring resonator (MRR)-based photonic vector modulators can be calibrated to achieve a root-mean-square (RMS) phase error of better than 2° and an amplitude error of 0.3 dB. Our findings identify a pathway to realize large-scale, fully-connected hybrid beamformers by leveraging compact and low loss photonic resonators.

Beamforming is a key enabling technology for wireless communication networks at higher frequency bands, offering increased capacity through spatial multiplexing and enhanced signal-to-noise ratio (SNR) due to the high array gain of massive MIMO[1]. Beamformers can be broadly categorized into one of three architectures: digital, analog, and hybrid. Analog beamformers (ABF) consist of a single transceiver chain connected to an antenna array through a network of splitters and phase shifters, which are configured to generate a desired array pattern. ABFs have low cost and power consumption, however, they generally do not support multi-beam operation, which is an essential feature to increase network capacity through multi-user MIMO[2]. Digital beamformers, on the other hand, have a full transceiver chain for each antenna in the system, bringing high performance and signal processing flexibility and allowing generation of multiple concurrent beams up to the number of antennas in the system. This comes at the expense of increased hardware complexity and power consumption, particularly in the millimeter wave (mm-wave) frequency bands (24.25 GHz–71.0 GHz), where the higher bandwidth requirements further drive up the power consumption of the transceiver chains.

Hybrid beamforming combines a low dimensional digital beamformer in the baseband domain with a high dimensional analog beamformer in the radio-frequency (RF) domain, striking a better balance between performance and power consumption than can be attained with a fully digital or fully analog solution[3]. Hybrid beamformers may be partially connected (PC), consisting of multiple subarrays with independent transceiver chains, or fully-connected (FC), in which each transceiver chain is connected to every antenna element in the array[4]. PC architectures have low hardware complexity, while FC hybrid beamformers offer better spectral efficiency with maximum beamforming gain. A considerable challenge in realizing a fully-connected hybrid beamformer at higher frequency bands is the complexity of the feed network for the analog part of the beamformer. The signals from each transceiver chain must be split, phase shifted, and summed at each of the antenna elements, requiring a network of splitters, couplers, crossings, and phase shifters that generally have high insertion losses and large footprints in conventional microwave circuits[5]. As a result, mm-wave base stations usually employ partially-connected architectures to reduce the hardware complexity and power consumption of the system, sacrificing coverage for spatial multiplexing capabilities as shown in Fig. 1.

[1]University of British Columbia, Vancouver, BC, Canada. [2]Queen's University, Kingston, ON, Canada. [3]Huawei Technologies, Markham, ON, Canada. ✉e-mail: m.reed.nichols@gmail.com

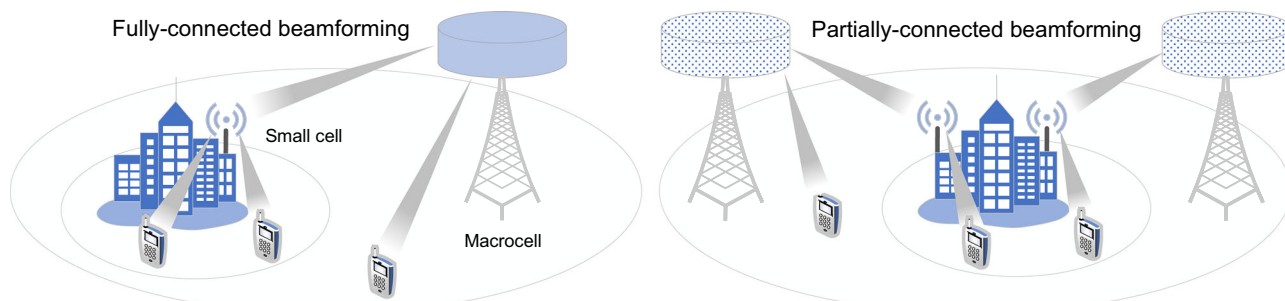

**Fig. 1 | 5G wireless network with fully-connected and partially connected beamformers.** The loss of array gain incurred when using partially-connected beamforming may necessitate additional cells to service the same coverage area, thereby increasing the deployment costs of the network in comparison to fully-connected architectures, which offer spatial multiplexing features without compromising the array gain.

Recently, photonic implementations of multi-beam phased arrays have been explored as a potential solution to the limitations of conventional microwave electronics for fully connected hybrid beamformers[6,7]. Optical beamformers can be classified as either coherent or incoherent[7]. In coherent systems, the light source for each signal path is derived from a single laser with power combining performed in the optical domain. This approach requires precise phase control to preserve coherence within the feed network, which may pose a challenge due to the harsh environmental conditions that a wireless base station must tolerate. Incoherent architectures generally use a multi-wavelength source with each input signal assigned to a different carrier wavelength. The modulated optical signals are multiplexed onto a single waveguide, and power combining is realized through the mixing of photocurrents in the electrical domain, after photodetection. Multiple laser sources are required, although the power requirements for each source are relaxed, as is the need to maintain precise phase control through the feed network, since combining is realized in the electrical domain where the signal wavelengths are much larger. Earlier work on incoherent optical beamformers has used true time delays in place of phase shifters to mitigate the beam squint effect that can limit performance in wideband systems[8]. However, delay lines consume large chip areas and generally incur higher insertion losses than phase shifters, even in photonic integrated circuits, making this approach unfavorable for massive MIMO systems where hundreds or thousands of delay lines are required. Incoherent optical beamforming based on phase shifters could therefore see applications in 5GB mm-wave networks. Previously, a microring resonator (MRR)-based weight bank has been proposed for weighted addition of a wavelength-division multiplexed (WDM) input vector[9,10]. This optical signal processor leverages the complementary outputs of a double-bus ring resonator with differential detection to perform signed multiplication with an input signal at a particular wavelength by shifting the resonance peak of the ring relative to the wavelength of the input signal. A dot product of arbitrary length can therefore be implemented by cascading multiple ring resonators in parallel with resonant peaks at different wavelengths[11]. The intensity of each wavelength channel represents the elements of the input vector, while the modulating signal applied to the ring resonators represents the elements of the weight vector. MRR weight banks are highly compact in comparison to other optical signal processing structures, such as Mach-Zehnder interferometer (MZI) mesh networks, enabling dense integration of large-scale matrix multiplications in photonic integrated circuits.

The MRR weight bank has found applications in blind interference cancellation for cognitive radio[12], nonlinearity compensation in long-haul fiber-optic transmission[13], and as a photonic neuron in an optical neural network[14]. In this paper, we build on our previous work[15], extending our photonic vector modulator to a multi-beam phased array that overcomes the challenges of fully-connected hybrid beamforming by leveraging MRR weight banks to provide dense, weighted interconnections between the RF chains and antenna elements. We report the results of a small-scale experimental beamforming demonstration that highlights the feasibility of our architecture and shows that RF phase shifters implemented by MRR weight banks can be calibrated to achieve high phase accuracy while remaining relatively insensitive to inter-channel crosstalk. We discuss the scalability of this architecture at high radio frequencies for massive MIMO systems and present a modified MRR weight bank structure for coherent RF signal processing with two key innovations. First, all-pass filters (APFs) are used to eliminate the weight-dependent delay through the add-drop filters, ensuring that the group delay can be matched across all wavelength channels. Second, an additional passive ring is inserted in series between each of the add-drop filters and the drop-port bus, which ensures that the waveguide lengths can be matched for all paths through the circuit. Our results represent a proof-of-concept of our proposed optical beamforming system and provide motivation for future work in experimental beamforming demonstrations using MRR weight banks at massive or extreme MIMO scales.

## Results

### System architecture

Cartesian phase generation is a technique to implement a phase shifter by independently weighting the in-phase and quadrature components of the input signal. The benefit to this method is that delay lines, which generally occupy a large area and introduce high insertion losses, are not required. Furthermore, the amplitude of the input signal can be adjusted in addition to the phase, which can be useful in some beamforming applications to taper the weights for sidelobe suppression[16]. A vector modulator is an example of a Cartesian phase shifter based on variable gain amplifiers or attenuators and a quadrature splitter or combiner. Analog phased arrays can be implemented using a bank of parallel vector modulators with a splitter or combiner network at the input, as shown in Fig. 2a[17].

Exploiting linearity in the signal chain, we can commute the summation and 90° phase shift operations such that the in-phase and quadrature components of the input signal are weighted and combined separately before the 90° shift, as shown in Fig. 2b. Therefore, the vector-modulator-based phased array can be expressed as two dot product operations for each of the in phase and quadrature components of the signal, followed by a quadrature coupler. An electronic implementation as described in ref. 18 consists of paired programmable-gain amplifiers on each antenna chain to implement RF domain complex-valued Cartesian combining followed by a quadrature downconversion stage. This electronic approach can efficiently implement a two-beam phased array that scales well with the number of antenna elements. However, increasing the number of RF chains in the architecture is non-trivial and would necessitate waveguide crossings to distribute the antenna signals across each of the chains, incurring large insertion losses and interchannel crosstalk.

We developed an incoherent optical beamforming architecture using phase shifters based on photonic in-phase and quadrature (IQ) vector modulators (IQVMs) with the aim to overcome the scaling challenges of electronic hardware and the limitations of existing optical solutions based on true time delays. Our solution leverages MRR weight banks to implement the weighted combiners in the Cartesian phase generation stage of the

**Fig. 2 | IQ-based phased array architecture using vector modulators. a** Phased array based on in-phase and quadrature (IQ) vector modulators (IQVMs). **b** IQVM phased array expressed as a weighted combination of the input signals by a quadrature coupler. The inputs $x_i$, $i = 1, \ldots, N$ represent the signals received at an antenna array, while the output, $y$, can be expressed as a dot product of the input signals with the complex weights $w_{k,i} + j w_{k,j}$, $k = 1, \ldots, N$.

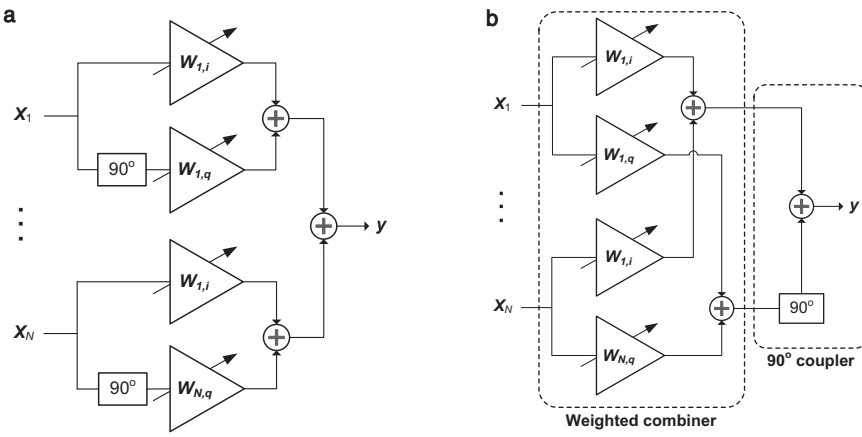

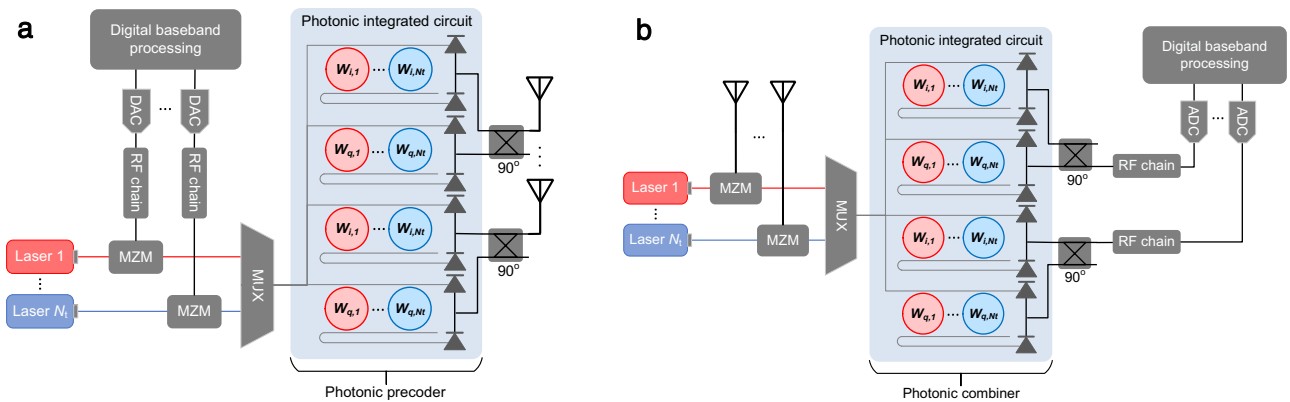

**Fig. 3 | Fully-connected photonic hybrid beamforming system with microring-resonator-based RF phase shifters. a** Transmitter configuration. **b** Receiver configuration.

beamformer, followed by an RF hybrid coupler to recombine the signal quadratures as illustrated in Fig. 3. In a transmitter scenario, the signals from a digital baseband processor are first converted to an analog representation using digital-to-analog converters (DACs) before RF domain processing including frequency upconversion, filtering, and amplification. The signals from the $N_t$ RF chains are modulated onto a set of optical carriers using a Mach-Zehnder modulator (MZM), and multiplexed (MUX) onto a single waveguide using WDM. The WDM signal is split and broadcast to a total of $2N$ MRR weight banks, one for each of the I and Q components and for each of the $N$ antennas. After photodetection, the in-phase and quadrature (I and Q) signal pairs are combined through a 90° hybrid coupler to form the output signals from the phased array, consisting of a superposition of the phased shifted signals from each of the RF chains. In a receiver configuration, a total of $N$ optical carriers (equal to the number of antennas) are modulated by the signals received at an antenna array that contain multiple components with different angles of arrival. The WDM signal is split into $2N_t$ parallel paths and injected into a set of MRR weight banks that implement the I and Q channel weights for each of the beams in the phased array. Here, $N_t$ is the number of transceiver chains in the receiver. The I and Q signals are then recombined through a hybrid coupler before RF processing and finally digitization through the analog-to-digital converters (ADCs). This approach, inspired by the broadcast-and-weight protocol proposed by Tait et al.[19], offers a solution to the signal distribution problem in a large scale multi-beam phased array. The use of WDM in the feed network significantly reduces the footprint of the system and eliminates the need for waveguide crossings to route signals between the transceiver chains and antennas.

Alternatively, this may also be accomplished by repartitioning the system in Fig. 3 so that the MRRs are used as simple de-multiplexers with phase shifting performed in the electrical domain. This approach would simplify the calibration and control of the ring resonators and may improve

the end-to-end insertion loss of the beamformer, depending on the gain of the RF phase shifters; however, it would come at the cost of significantly increased complexity in the electrical interfaces. Each MRR would require a dedicated photodetector, which would contribute considerable size and power to the overall system, as there would be a total of $2N$ or $2K$ detectors, depending on whether the system is a receiver or transmitter. Furthermore, this solution would require an RF phase shifter per antenna and per beam, which would incur an enormous size and penalty cost in multi-user massive MIMO systems where the number of beams and antennas is large. By leveraging MRR weight banks as phase shifters, our proposed hybrid optical beamforming system provides an efficient solution to both signal distribution and processing for scaling up next-generation massive MIMO systems.

## Experiment setup

To evaluate the feasibility of our proposed optical beamforming architecture, we conducted an experimental optical beamforming demonstration with IQ vector modulator phase shifters based on MRR weight banks. The experiment was designed to emulate a two-element receive-mode phased array using an arbitrary waveform generator (AWG) to simulate a set of 1 GHz RF signals received by two half-wavelength spaced antennas at various angles of arrival. Each signal generated by the AWG is connected to a 90° splitter, amplified, and injected into the arms of a lithium niobate (LiN) dual-drive intensity modulator biased at the quadrature point to produce a single sideband signal on each of the two wavelength channels at 1555.34 nm and 1553.73 nm. The RF-modulated signals are then multiplexed onto a single fiber using an arrayed waveguide grating with a 50 GHz channel spacing and amplified to 13 dBm by an erbium doped fiber amplifier before injection into a photonic integrated circuit (PIC). The PIC contains two MRR weight banks with balanced photodetectors, each weight bank supporting up to four wavelength channels. The optical signals are

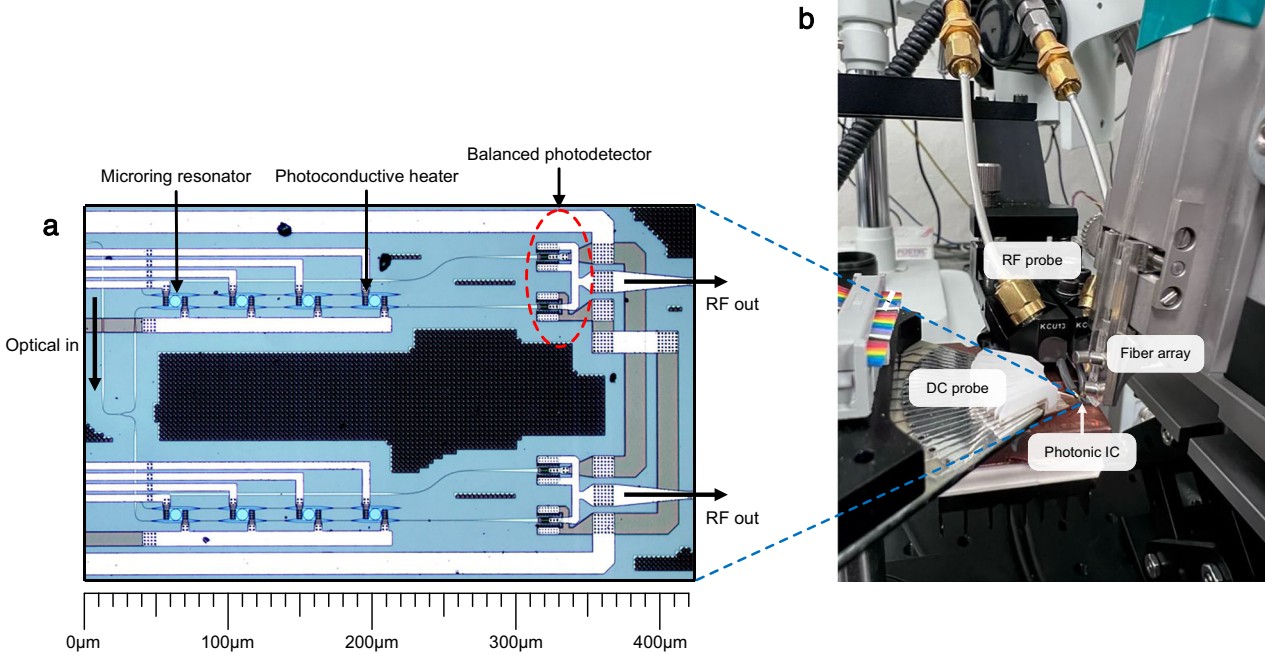

**Fig. 4 | Photonic beamforming chip and measurement setup. a** Micrograph of the on-chip ring resonators and balanced photodetectors used in the demonstration. **b** Optical beamforming experiment setup with DC, RF, and optical probes aligned to the photonic integrated circuit.

coupled from the fiber to the chip using an optical V-groove manually aligned to the on-chip grating couplers. The weight banks are controlled using a source measure unit (SMU) with a DC probe connected to electrical pads on the PIC. After weighting and detection, the RF signals at the balanced photodetector outputs are brought off chip using a ground-signal-ground-signal-ground (GSGSG) probe and recombined in quadrature through another 90° hybrid coupler and sampled for offline calibration. The on-chip microring weight banks used in this experiment are shown in the micrograph of Fig. 4a, while Fig. 4b depicts the optical beamforming experiment setup with the inputs, outputs, and control probes aligned to the PIC.

**Characterization of phase shifter accuracy**
Programming the MRR weight banks requires a calibration and control procedure to determine the current or voltage signals required to actuate the rings, given the set of desired weight values. Previously, feedback control of MRR weight banks has been demonstrated by sensing the photoresponse generated by in-ring n-doped photoconductive heaters[20]. This method achieved 5–7 bits of precision in the weight control; however, it requires precise measurement of the voltage across the heater to sense the change in resistance caused by the light circulating within the ring. The peak $\Delta V$ of the photoresponse is small relative to the total voltage across the heater, and would therefore require a high precision ADC to achieve a satisfactory weight precision. This would increase the cost and power consumption of an on-chip implementation that may outweigh the advantages provided by a photonic signal processor over the electronics-based alternatives. Instead, we use a feedforward calibration method and demonstrate that the crosstalk between weight elements has a tolerable impact on the root-mean-square (RMS) phase error of the phase shifter implemented by the MRR weight bank.

We calibrated the two MRR IQ vector modulators with 5 bits of precision by building a look-up table (LUT) mapping the desired phase shift to a pair of heater currents for each of the I and Q channels of the vector modulator. The accuracy of the phase shifters was then measured in isolation and in the presence of thermal crosstalk between the two channels. The quality of the phase shifters was evaluated by measuring the amplitude and phase of the output RF signal for each of the $n_{ps} = 32$ ($2^5$) target phase values and computing the RMS amplitude and phase error. After calibration, the

measured RMS amplitude and phase error for the two MRR IQ vector modulators in this experiment were 1.84°, 0.22 dB and 1.20°, 0.21 dB, which is competitive with high-performing analog electronic solutions[21,22]. The calibrated complex constellations for each of the two input channels are shown in Fig. 5a.

Next, we evaluated the system performance subject to spectral and thermal crosstalk. To achieve this, we performed a two-dimensional sweep over all achievable phases in both of the MRR IQVM phase shifters. The results are depicted in Fig. 5b. Comparing to Fig. 5a, we can see that the accuracy of the calibrated weights is slightly degraded. Furthermore, there is small drift factor as the phase is adjusted in the adjacent MRR IQVM, most likely due to thermal and/or spectral crosstalk between the rings. However, the RMS phase and amplitude errors remain below 4° and 0.3 dB across all weight settings, which is within an acceptable range for beamforming applications[23,24]. The results demonstrate that our proposed MRR IQVM is resilient to spectral and thermal crosstalk between the ring resonators and may allow a simple feedforward control scheme to be used rather than the higher complexity feedback approaches schemes that have been demonstrated in previous works. In future work, the feedforward control rule may be extended to operate over a wide temperature range to relax the thermal stability requirements.

**Array pattern synthesis**
Finally, we assessed the performance of the calibrated system in a realistic beamforming application. The MRR IQVM beamformer was configured with a fixed pointing direction, while the relative phase shift between the input RF signals generated by the AWG was swept from $[-\pi, \pi]$ to simulate an input signal with arrival angles of $-\pi/2, \pi/2$. The sweep was repeated for multiple configurations of the MRR weight bank corresponding to arrival angles of $-60°, -30°, 0°, 30°, 60°$. The output signal power was measured and plotted against the virtual angle-of-arrival to reconstruct the array pattern of the beamformer. The results in Fig. 6b show, as expected, the signal power is maximized when the arrival angle coincides with the pointing angle of the beamformer so that the input signals are summed coherently. The ideal two-element array pattern generated by simulation with ideal 5-bit phase shifters is shown for reference in Fig. 6a. The measured array pattern in this experiment is closely matched to the ideal result; however, the peak-to-null ratio is low at approximately 8 dB. This may be

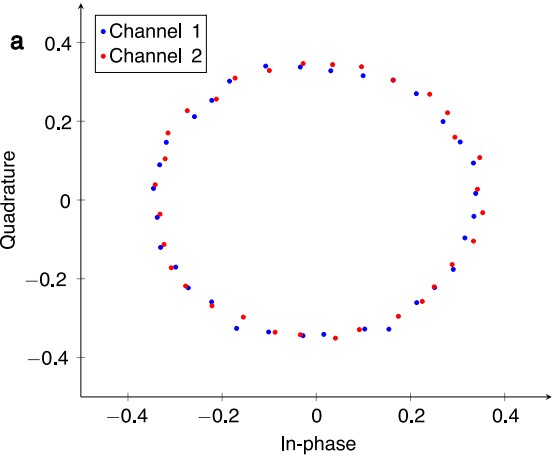 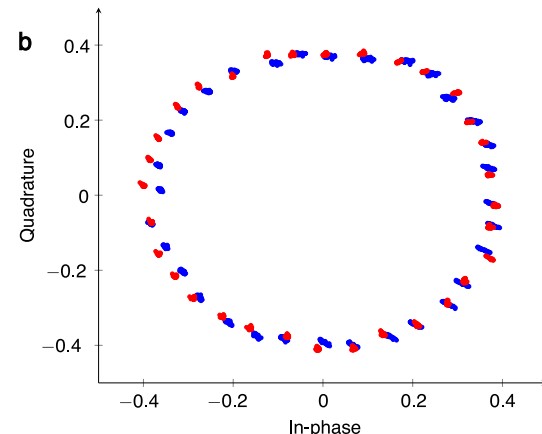

**Fig. 5 | Constellations from calibrated phase sweeps in microring resonator vector modulators. a** Complex weight constellation of the two calibrated microring-resonator-based phase shifters demonstrated in this experiment. **b** Constellation of all $n_{ps}^2$ weight pairs generated by a sweep over the $n_{ps}$ calibrated phase shift values for each of the two phase shifters.

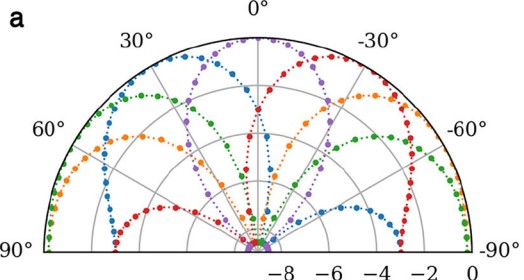 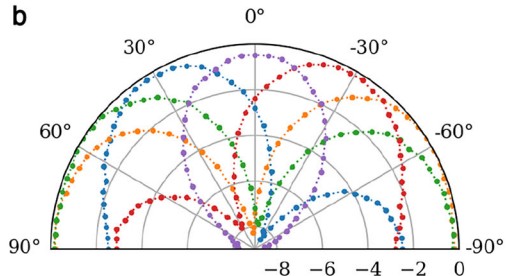

**Fig. 6 | Ideal and measured beam patterns of a two-element phased array.** Beamformed signal power on a normalized dB scale vs. angle-of-arrival of the input signal. Each curve represents a sweep of the input signal arrival angle with a fixed pointing angle in the beamformer. **a** Ideal two-element array pattern at various angles of arrival on a normalized dB scale and **b** measured array pattern generated with the experimental optical beamformer.

explained by signal impairments in the weight bank due to mismatched delays through the system, which we can be addressed by modifying the topology of the weight bank, as we will discuss in the following section.

## Discussion

In this experiment, we set out to show that a pair of microring weight banks can be configured as a highly accurate RF phase shifter for beamforming applications. The results highlight the feasibility of our proposed optical beamforming architecture and provide motivation for future work in fully-connected optical hybrid beamforming in the mm-wave bands (<28 GHz). Given the compact form factor of microring resonators, this approach could offer a substantial footprint reduction in comparison to previous incoherent optical beamformers, based on true time delays or dispersive fibers. However, there are challenges that must be addressed to facilitate the deployment of an MRR IQVM optical beamformer in a massive MIMO system with 32 antennas or more. Here, we identify those challenges and present our solutions.

### Challenges and design considerations

The first challenge is the optical waveguide technology, which is a key consideration for the design of large-scale systems. For example, silicon nitride (SiN) technology offers extremely low propagation losses; however, the low index contrast reduces the confinement in SiN waveguides, which increases the bending loss at the small radius of curvature required for high-bandwidth microring weight banks. A silicon-on-insulator (SOI) platform offers higher index contrast, which reduces the losses in small bends. However, SOI waveguides are prone to scattering losses due to the roughness of the sidewalls. Furthermore, the high Kerr nonlinearity and two-

photon absorption in SOI limit the maximum optical power to the tens of milliwatts range. For high dynamic range links, the required optical power at the detector is on the order of milliwatts[25], and the total accumulated losses can exceed 20 dB after accounting for the splitter network, modulator, and weight bank, pushing the total input power requirements beyond the power handling limits of SOI[26]. Therefore, a practical implementation may require a hybrid or heterogeneous integration platform with the distribution network implemented in SiN waveguides, where high-power handling is required, and the MRR weight banks in a SOI waveguide system, where tight bends are needed. Integration with active devices based on a III–V semiconductor platform, such as indium phosphide (InP), is also desirable, so that optical amplifiers can be embedded in the circuit to compensate for insertion loss in the modulators and distribution network[27].

The microring resonators must be carefully designed to minimize the insertion loss through the rings and to ensure that all paths through the routing network are delay-matched. The coupling condition of the rings is an important factor to consider that affects the relative delays between the through and drop ports. The critical coupling condition must be satisfied to ensure that the signals exiting the ring at the through and drop ports are delay-matched. This is achieved by matching the losses in the couplers to the loss in the ring waveguide as given by the equation $r_1 = r_2 a$, where $r_1$, $r_2$ are the through and drop port coupling coefficients and $a$ is the round trip loss coefficient[28]. Designing the Q factor of the MRRs presents trade-offs in the balance of loss and bandwidth of the filters. A high Q factor is desirable to minimize signal loss within the rings and maximize the free spectral range; however, the full-width half-maximum (FWHM) must be sufficiently wide to pass the modulated optical signal without attenuating the sideband. This penalty can be seen by considering the drop port transmission of an add-

drop ring resonator as a function of the detuning factor $\phi$:

$$T_d(\phi) = \frac{(1-r_1)^2(1-r_2)^2 a}{1 - 2r_1r_2a\cos(\phi) + (r_1r_2a)^2} \qquad (1)$$

At high RF frequencies, the signal bandwidth is small relative to the carrier frequency; therefore, we can approximate the single sideband modulated optical signal as a dual tone signal with a spacing of $f_{RF}$ between the carrier and sideband. Under this assumption, it can be seen that the add-drop filter attenuates the drop port signal by a factor of $T_d(\phi)$, where $\phi = 2\pi f_{RF}/FSR$ and FSR is the free-spectral range of the ring in Hz. The insertion loss of the MRRs can therefore be mitigated by maximizing the FWHM of the ring subject to a minimum finesse constraint to minimize the detuning loss through the drop port of the filters while maintaining a sufficiently large FSR to accommodate the desired number of wavelength channels. The maximum channel count of the MRR weight bank can be expressed as $N \leq \frac{F}{\delta\omega}$ where $F$ is the finesse of the rings and $\delta\omega$ represents the channel spacing normalized to the resonator FWHM[10]. In previous works, the crosstalk-limited normalized channel spacing has been estimated in the range 3.1–8.8[10,19]. Taking a pessimistic assumption of 10 linewidths minimum channel spacing, the ring resonators would require a finesse of at least 320 and 640 to support massive MIMO configurations of 32T32R and 64T64R. It is therefore challenging to scale the proposed beamforming architecture up to ultra-high frequencies in the high mm-wave and THz bands while retaining high channel counts due to the bandwidth requirements that would be imposed on the rings. Architectural changes may need to be considered to extend the frequency of operation beyond the low mm-wave bands. For example, two-pole MRR weight banks can support greater channel densities than ordinary single-pole MRRs[11], however, this would increase the footprint of the filters and the complexity of the control electronics.

The large number of cascaded ring resonators and wavelength channels in an MRR IQVM receiver also poses a challenge for system packaging and integration. Feedback control may be required to stabilize the ring resonances in the presence of temperature drift, which would increase the cost and power consumption of the system substantially due to the number of ADCs and DACs needed to monitor and actuate the weights. There is therefore a strong motivation for feedforward control, which would eliminate the need for ADCs in the control circuits to monitor the resonance condition of the rings. The issue of thermal instability that plagues silicon microring resonators would need to be addressed by a passive solution; for example, by embedding negative thermo-optic materials in the rings, or using thermally-matched interferometric structures[29]. It should also be noted that process variations must be corrected to ensure that the ring resonances are aligned to the wavelength channels. This can be accomplished using the tuning elements within the rings, provided that they have sufficiently wide tuning range, or by post-fabrication trimming of the ring resonances[30].

## Scaling up for mm-wave massive MIMO

The scalability challenges in fully-connected beamforming networks arise from the complexity of the signal distribution network and the large number of phase shifters, both of which grow rapidly with increasing antenna counts and spatial channels. Leveraging WDM in the feed network directly addresses these challenges by encoding multiple parallel signal paths into distinct wavelength channels. Although this approach necessitates a dedicated laser source for each input, the tradeoff is favorable as the MRR weight banks used for phase shifting are extremely compact, particularly compared to conventional phase shifters. Given the large number of phase-shifting elements required in a fully-connected beamformer, this compactness substantially offsets the additional complexity introduced by multiple lasers. For instance, consider a 32 element beamformer with 4 spatial channels in a receiver configuration. The lasers dominate the system area as dense integration is challenging, and the number of wavelength channels is equal to the number of antennas. However, state-of-the-art integrated lasers have been demonstrated with a footprint as small as 0.5 mm². Similarly, recent advances in integrated photonics have yielded ring-assisted Mach-Zehnder modulators with a chip area of only 0.005 mm²[31], and wavelength-division-(de)multiplexers with a footprint of approximately 0.5 mm²[32]. Assuming a ring diameter of 5 μm, and considering there are two MRR weight banks per spatial channel, the total area of the beamformer can estimated as $A_{OBF} = 32 \cdot A_{MZM} + 32 \cdot 4 \cdot 2 \cdot A_{MRR} + 32 \cdot A_{Laser} + A_{WDM} = 16.67$ mm². The form factor can be further reduced by leveraging optical frequency combs as a multi-wavelength laser source to replace the discrete lasers, however, this approach would necessitate careful calibration of the frequency comb to match the power levels of each wavelength channel.

There are two issues that arise when using MRR weight banks for RF signal processing that limit the scalability of the system in large antenna arrays. The first stems from the fact that the through and drop ports of the microring weight bank propagate in opposite directions, with the drop port bus looped back around to the detector. This path length mismatch can induce a considerable delay difference between the through and drop port signals that would lead to incoherent subtraction at the balanced photodetector, even when the rings are designed to satisfy the critical coupling condition. Even-order weight banks (two-pole, four-pole, etc.) do not suffer this issue as the through and drop ports propagate in the same direction. However, higher order filters have a steeper roll-off and a narrower FWHM, which would limit the optical bandwidth of the weight bank[11]. The second limitation of the MRR weight bank RF signal processing approach is related to the group delay of the add-drop filters, which induces a weight-dependent phase variation as described previously.

**Path length mismatch.** Figure 7a illustrates the issue of path length mismatch between the through and drop ports. It can be seen that the path length from each ring to the detector is longer along the drop port bus than the through port bus. This is due to fact that the signal exiting the ring at the drop port propagates in the opposite direction of the signal exiting at the through port. Note that the severity of the path mismatch varies across the WDM channels depending on the position of the corresponding ring resonator within the weight bank. For example, the ring farthest from the balanced photodetector sees nearly equal paths to the detector from the through and drop ports, with only a small mismatch

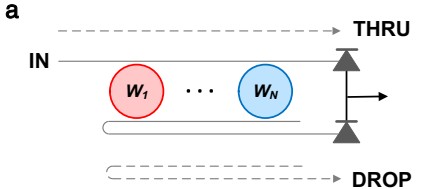

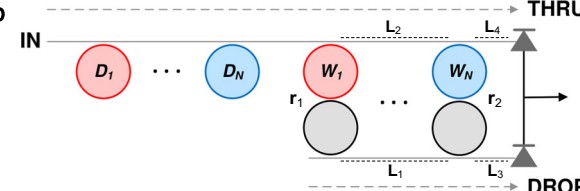

**Fig. 7 | Precompensation of through-drop path-length mismatch in a microring weight bank. a** Illustration of the path length mismatch between the through and drop ports in an ordinary microring weight bank. **b** RF coherent microring weight bank with all-pass filter (APF) precompensators to equalize the delay through the weight elements and passive delay rings coupled to the drop port to reverse the direction of signal propagation in the drop port bus waveguide. Here $W_1, \ldots, W_N$ represent the weights applied in the add-drop filters (ADFs), and $D_1, \ldots, D_N$ denote the weights in the APFs which are tuned to compensate the delay of the ADFs while maximizing the transmittance of the ring to avoid perturbing the signal amplitudes. $L_1 - L_4$ are the lengths of the interconnecting waveguides within the weight bank and to the detector.

from the half loop required to route the backward propagating drop port signal forward to the detector. On the other hand, the resonator closest to the detector has a severe mismatch in path length as the through port connects directly to the detector, while the drop port is routed back through all the previous rings. Considering a ring radius of 5 µm, the length mismatch increases by a minimum of 20 µm for each additional ring in the weight bank. Assuming an approximate group velocity of $1.2 \times 10^8$ m/s, then in the worst case, each ring induces a 1.8° phase shift between the inverting and non-inverting inputs of the balanced photodetector at an RF carrier of 30 GHz without accounting for any thermal isolation spacing.

To resolve this issue, we propose a modified weight bank structure with a set of passive rings connected in series to the ADF weight elements with full power coupling to the drop port waveguide. In this configuration, the passive ring behaves as a pure delay while serving to reverse the direction of the drop port so the through and drop port signals propagate in the same direction towards the photodetector. The signals exiting the drop port of the ring connected to the through port bus are coupled into the second ring, traverse a half-loop of the ring, and are then coupled to the drop port bus with 100% efficiency. The signals then propagate along the drop port bus, traversing one full circuit of the lower ring for each of the subsequent weight elements. The signal path along the drop port is therefore longer than that of the through port, however, the paths can be matched by lengthening the interconnecting waveguides between the through port and input port of each ring to account for the additional delay through the passive rings at the drop port.

Figure 7b shows how the signal paths can be matched using this weight bank structure. The signal exiting the drop port of the rightmost ring propagates a total distance of $2\pi r_2 + L_3$, where $r_2$ is the radius of the rightmost ring, and $L_3$ is the path length from the rightmost ring to the detector. The signal exiting the through port travels a total distance of $L_4$, so the delays can be matched by setting $L_4 = L_3 + 2\pi r_2$. Similarly, the signal exiting the drop port of the leftmost ring travels a total distance of $2\pi r_1 + L_1 + 2\pi r_2 + L_3$, where $r_1$ is the radius of the leftmost ring and $L_1$ is the length of the waveguide connecting the rings along the drop port bus. The signal exiting the through port of the leftmost ring propagates a total distance of $L_2 + L_4$, so the path lengths along the through and drop ports are matched when $L_2 = L_1 + 2\pi r_1$. This scheme works with any number of rings in the weight bank. The through and drop port delays will be matched provided that the length of the through port waveguide connecting any pair of rings is equal to the circumference of the first ring plus the length of the drop port waveguide connecting the ring pair.

**Weight-dependent group delay variation.** In addition to the variable propagation delays through the weight bank, the group delay variation of

the rings themselves must be addressed to ensure that the amplitude of the signals can be weighted without affecting their phase relationship. To overcome this issue, a bank of all-pass filters can be placed before the weight elements to precompensate the residual phase modulation of the ADFs in the MRR weight bank. Figure 8a shows that the phase response of all-pass and add-drop filters is similar along one side of the resonance peak when the coupling ratios are configured so that the FWHM of the rings is matched. When this condition is satisfied, the APF can be tuned to complement the delay in the ADF so that the overall phase response of the system is flat, as shown in Fig. 8b.

If an ADF weight is tuned on-resonance, for example, then the corresponding wavelength would experience a large delay, so the APF compensator is tuned off-resonance to minimize the cascaded delay through the weight bank. Conversely, if an ADF is tuned off-resonance to its corresponding wavelength channel, then the APF is tuned on-resonance to ensure that the delay through the cascaded APF and ADF pair is constant regardless of the weight bank configuration. This approach ensures that the phase relationship between the input signals is preserved after propagating through the MRR weight bank. We simulated the RF coherent MRR weight bank in a photonic vector modulator and confirmed that this new weight bank architecture eliminates impairments in the complex weight constellation that occurs at high RF frequencies. The results, depicted in Fig. 9a, show that for a 25 GHz input signal, the complex weight constellation of the MRR IQ vector modulator is severely distorted due to a combination of the mismatch between the through and drop ports, as well as the varying group delay through the ADFs. In Fig. 9b, we see that by modifying the photonic vector modulator to use RF coherent MRR weight banks, the distortion in the complex weight constellation is eliminated, aside from a constant rotation which can be easily calibrated at the system level. These results demonstrate that the APF precompensators combined with the passive delay rings overcome the limitations of ordinary MRR weight banks for high-frequency RF signal processing, which highlights the potential for the RF coherent MRR weight bank to realize ultra-compact high-dimensional analog beamformers for hybrid beamforming applications.

## Conclusion

In this work, we proposed a new optical beamforming architecture based on tunable optical filters that addresses the implementation challenges of fully-connected hybrid beamformers on electronic platforms. We performed an optical beamforming experiment with a two-element phased array using microring-weight-bank-based photonic vector modulators and demonstrated a post-calibration phase accuracy better than 4° RMS. The results

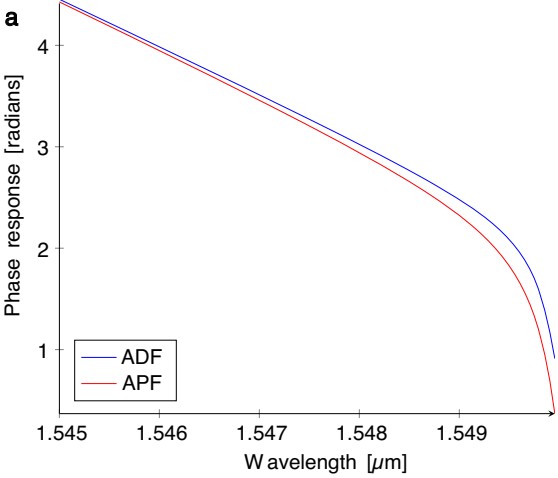
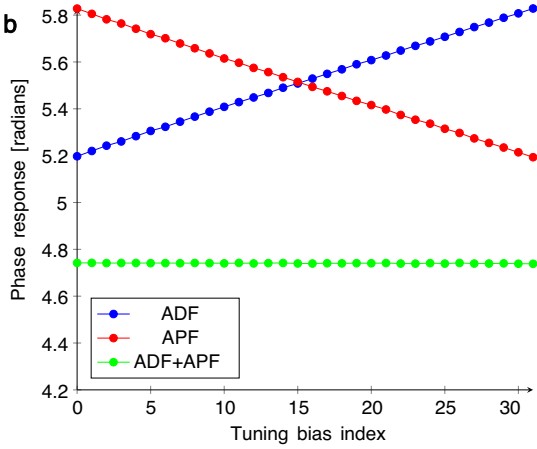
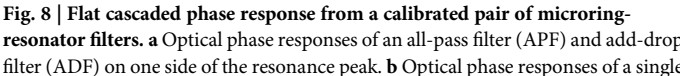

**Fig. 8 | Flat cascaded phase response from a calibrated pair of microring-resonator filters. a** Optical phase responses of an all-pass filter (APF) and add-drop filter (ADF) on one side of the resonance peak. **b** Optical phase responses of a single APF and ADF calibrated to provide a complementary phase shift at every weight setting so that the overall phase response of the cascaded APF and ADF is flat.

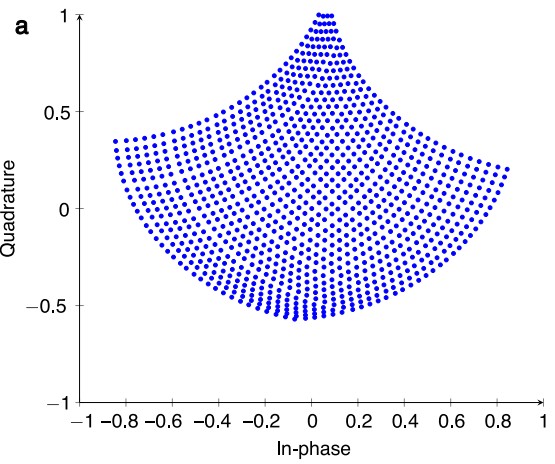

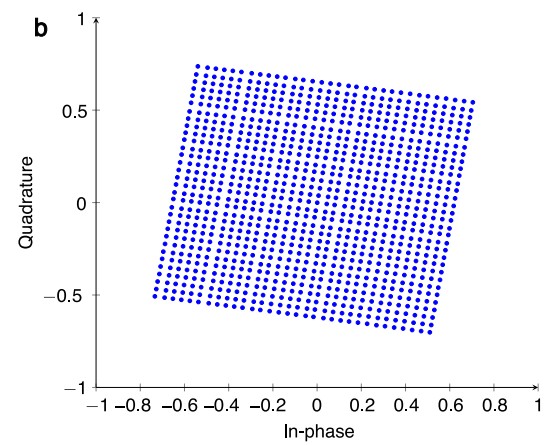

**Fig. 9 | Simulated complex weight constellations from microring resonator vector modulators.** Simulated complex weight constellation for a microring resonator (MRR) vector modulator with a 25 GHz input signal using **a** generic MRR weight banks and **b** an RF coherent MRR weight bank. The constellations are generated by sweeping the I and Q channel add-drop filter (ADF) weights over the tuning range of the resonator and uniformly quantizing the weight range with 5 bits of precision. For the RF coherent MRR weight bank, an additional step is performed to sweep the tuning bias in the all-pass filter (APF) precompensator. Next, a bias point for the APF is selected for each of the 32 ADF weights so that the cascaded delay through the APF and ADF is constant across all weight settings.

highlighted robustness of MRR IQ vector modulators in the presence of hardware nonidealities such as thermal and spectral crosstalk. We discussed the prospect of scaling this beamforming architecture up for mm-wave massive MIMO systems and proposed a modified MRR weight bank for high-dimensional, high frequency RF signal processing, which was simulated and shown to address the issue of path-dependent delay variations in generic MRR weight banks. Future work will focus on technologies and methods to further extend the scalability of an MRR IQVM-based receiver, which faces challenges in meeting massive MIMO system requirements due to the large number of wavelength channels that would be needed. Potential solutions to these challenges include mode-division multiplexing, which could be used as a complement to WDM to increase the maximum number of channels, optical frequency combs, which could be explored to mitigate the complexity of integrating many wavelength sources into the system, and FSR-free optical resonators such as photonic crystal nanobeam cavities, which could enable weight banks with large bandwidths and channel counts[33,34].

## Methods
### Calibration and control procedure
The goal of the calibration procedure is to find a mapping between the target phase values and the tuning signals required to actuate the rings to achieve the target phase shifts. The procedure consists of three steps. First, an initial sweep is performed to acquire a high resolution mapping between the tuning currents and the realized weights in each ring. The current is swept over a large current range (up to 0.8 mA) with high precision (7–8 bits) while measuring the RMS voltage of the output RF signal at each step. This sweep is repeated for each of the I and Q rings and for each of the wavelength channels. Only one laser is active at a given time, corresponding to the wavelength channel of the ring being actuated. At the next step, the current sweep for each ring is resampled to linearly quantize the measured RMS voltage of the RF signal within the range $[-W_{max}, W_{max}]$ with 5 bits of precision, where $W_{max}$ is determined by the ring with the smallest positive or negative weight bound, so that all of the rings in the bank can be calibrated to the same weight range. Next, a two dimensional sweep is performed over the calibrated weights for each wavelength channel while measuring the RF signal at the output of the hybrid coupler. The measured signal is processed offline to extract the amplitude and phase information of the RF signal and form a complex constellation of the realized I and Q weights. Finally, a look-up table is constructed to map the required tuning currents in the I and Q rings to each of the 32 target phase shifts.

### Ring resonator coupling condition
The phase response of the weight elements poses a challenge for RF signal processing using MRR weight banks due to the large group delay of a microring resonator near the resonance peak. The signals at the through and drop ports must be synchronized to ensure that the photocurrents are subtracted in phase, however ring resonators can induce a substantial delay between the through and drop port signals depending on the coupling condition of the ring, as shown in Fig. 10b. In order to maintain phase alignment of the through and drop signals at the output of the ADFs, the rings must satisfy the critical coupling condition, which occurs when the coupling ratios are matched to the losses within the resonator, i.e., $\sqrt{1 - K_2}\sigma = \sqrt{1 - K_1}$ where $K_1$, $K_2$ are the cross coupling coefficients at the through and drop ports of the ring and $\sigma$ is the round-trip loss in the resonator[28].

When this condition is satisfied, the group delay of the ADF at the through port and drop port are identical, except for a small region at the resonance peak, which can be avoided by shifting the tuning region as shown in Fig. 10a. By constraining the weights to lie within this tuning region, the group delay experienced by the signals exiting the ring at the through port can be matched to the delay of the signal exiting at the drop port. This ensures coherence between the RF signals traversing the through and drop port buses so that the phase relationship of the signals is retained upon subtraction at the output of the balanced photodetector.

### Chip fabrication and testing
The devices were fabricated on a silicon photonics process of Advanced Micro Foundry (AMF) in Singapore through the support of CMC Microsystems. This process uses on a silicon-on-insulator wafer with a silicon thickness of 220 nm and a buried oxide thickness of 2 μm. The bus waveguides have a width of 500 nm. The MRR arrays on the chip have a base radius of $R = 8$ μm with a radius difference of $\Delta R = 12.13$ nm between adjacent rings, and a symmetric coupling gap between the two bus waveguides of $g = 200$ nm. The N-doped MRRs have a measured Q-factor of roughly 5500, an FSR of around 12 nm, and a measured tuning efficiency of roughly 0.25 nm/mW. The on-chip photodiodes have a measured bandwidth of 16 GHz at −2V reverse-bias. We use grating couplers with an 8° polish angle, which achieve a measured loss of 6 dB/coupler.

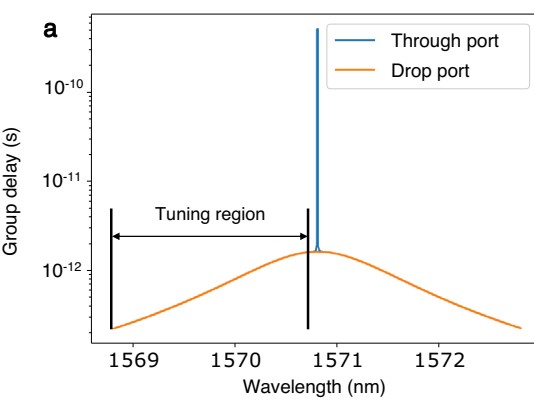
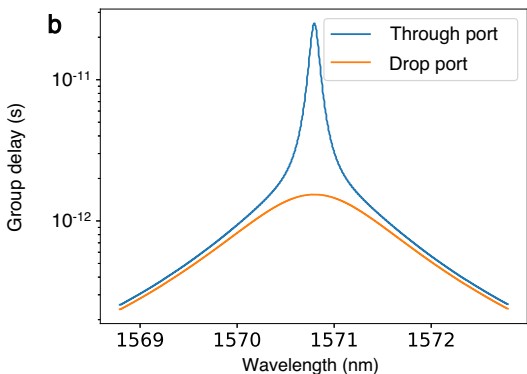

**Fig. 10 | Group-delay spectra at the through and drop ports of a microring resonator under critical coupling and overcoupling. a** Critical coupling. **b** Overcoupling.

## Data availability

All data supporting the findings of this study are included in this article.

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

## Author contributions

M.N. and L.L. conceived the idea. M.N. and H.M. designed and conducted the experiments. B.J.S. and A.E. contributed to the theoretical development of the work. All authors contributed to the manuscript. L.L. supervised the research.

## Competing interests

The authors declare the following competing interests: the research at the University of British Columbia (UBC) was funded through a research agreement with Huawei Technologies Canada Co., Ltd. M.N. held a paid internship at Huawei during part of the research period, and A.E. is an employee of Huawei. The industry partner provided nonbinding technical input on research directions. Two patent filings covering aspects of this work are co-owned by UBC, Huawei, and Queen's University.

## Patent disclosures

**US-20250023645-A1** - Applicant: Huawei Technologies Canada Co., Ltd.; The University of British Columbia; Queen's University at Kingston. Inventors: M.N., A.E., and L.L. Status: Pending. *Relevance to manuscript:* Fully connected optical beamforming system conceptualized and demonstrated in this work.
**US-11874501-B1** - Applicant: Huawei Technologies Canada Co., Ltd.; The University of British Columbia; Queen's University at Kingston. Inventors: M.N., A.E., and L.L. Status: Granted. *Relevance to manuscript:* Delay-compensating filter structure ensuring phase uniformity through all paths in the microring weight bank.
