## [Transparent Peer Review file · Communications Engineering]

Photonic Fully-Connected Hybrid Beamforming Using Mirroring Weight Banks

Corresponding Author: Mr Mitchell Nichols

Version 0:

Reviewer comments:

Reviewer #1

(Remarks to the Author)

This paper proposed an IQ-based RF phased-array architecture using previously demonstrated tunable photonic micro-ring resonators (MRR). While the application of MRR is already used for similar application (such as this reference: "Broadband physical layer cognitive radio with an integrated photonic processor for blind source separation", Nat Comm 2023), their use for phase-arrays can be an interesting topic for readers.

The paper overall is well-written and well-organized, however there are sever technical challenges and details that need to be addressed in this work:

1. I suggest avoid saying "... present a new approach to optical beamforming using microring weight banks as RF phase shifters ...". The MRRs are not actual RF phase-shifters and they only adjust the weights. So please make sure similar sentences are edited in next revision.
2. Details on photonic chip and devices need to be more complete. For instance, radius and Q-factor of MRRs, FSR, heater efficiency, Bandwidth of photodiodes, grating couplers loss, etc.
3. Authors should discuss the choice and trade-offs of MRR's Q-factors on the system performance. For instance, the Q-factor should be low-enough to accommodate the RF signal bandwidth, at the same time it impacts the MRR weights loss. Please include a section on design trade-offs of photonic devices specially MRRs.
4. In terms of overall practicality of proposed system (Fig. 3), one needs to consider all pros/cons vs. just distributing the signals optically and using electrical RF phase-shifters. For instance, total signal loss needs to be estimated and compares between these methods. (Notice that IQ signal amplitude is at 0.4 max!). Such an analysis needs to be included.
5. In the sections 3.1 and 3.2:
 - a. There's a discussion on SiN vs Si, and even use of amplifiers. This is misleading because no clear estimate of required optical/RF power is mentioned. Include some rough estimates for these, so that the readers can know the limitations/requirements of proposed approach.
 - b. Notice that while MRRs are thermally sensitive, it is essential (due to process variations) to have some tuning capability with large-range to initially adjust MRRs to laser lines. Make sure this is clear in your discussion.
 - c. The discussion on MRR's FWHM is incomplete because RF signal BW is not discussed. Also, number of wavelengths depends on FSR as well. Please edit these sections.
6. Which foundry has been used? There must be a chip fabrication and testing in Section 5 of Methods.

Reviewer #2

(Remarks to the Author)

This paper presents an integrated photonics architecture for hybrid beam forming based on microring weight banks, emphasizing scalability toward massive MIMO. It addresses an important issue and provides an interesting architecture. My comments are as follows:

Abstract

- Phase error of better than 2° — the degree symbol looks weird.

Results

- Figure 2.1 should be Figure 3?
- Feedforward control is interesting but needs more elaboration. What exactly is the rule and what can be extended?
- Fig 6 (a) and (b) are reversed with respect to the text description in section 2.4.

Discussion

- It's okay to compare the compact form factor of MRRs with other optical implementations using optical delay lines or dispersive fibers. However to compare with electrical solution, the entire optical system including the MZMs and the multiplexers should be included, both are quite large. Can the authors elaborate on this comparison and the scalability of the entire system, not only the MRR weight bank? Can comb-driven DWDM be used? Doesn't have to be multiple laser sources with a multiplexer.
- Scalability analyses: what limits the maximum N_t and N achievable? The analysis seems quite qualitative. At least some quantitative analysis using analytical models is desirable. Other factors limiting scalability could include power and control of the laser array, multiplexer and splitter design, MRR free-spectral range, accumulated loss on bus, etc, if the author could comment on as well.

Reviewer #3

(Remarks to the Author)

This paper proposes an approach to implementing an optical beamforming system based on photonic vector modulators using tunable photonic filters. The optical beamforming system is demonstrated by using micro-ring weight banks as RF phase shifters. The authors discussed the scalability of the architecture at high radio frequencies for massive MIMO systems and presented a modified MRR weight bank structure for coherent RF signal processing. This paper is timely and would draw significance to prospective readers provided the concerns are addressed satisfactorily.

Authors should discuss the significant advances that they have presented in the current manuscript as compared to Ref 1. Figure 10 of the current manuscript seems identical to Figure 3 of Ref. 1. The concept and physics of 'micro-ring resonator (MRR) weight banks' should be briefly introduced in the manuscript along with their advantages compared to other state-of-the-art photonic systems [2]. The operating principle of the proposed technique should be discussed as a sub-section for broad visibility and understanding of the technique to the readers. Authors should discuss the broad implications of their approach for photonic chip design and photonic signal processing; and how one can potentially incorporate artificial intelligence (AI) into the architecture to realize a more versatile and automated system. Given the authors discussed wireless beamforming, can authors comment on the polarization sensitivity of their approach? With a photonics-based approach, can authors realize polarization diversity and/or structured (polarization and phase-engineered) beamforming? A detailed discussion on this aspect is required. Authors should also comment on whether their architecture can provide beamforming for above 100 GHz+ frequencies.

References:

- [1] M. Nichols, M. Salmani, E. Luan, A. Eshaghi, and L. Lampe, "Photonic Vector Modulator Based on MRR Weight Banks," in Conference on Lasers and Electro-Optics, Technical Digest Series (Optica Publishing Group, 2022), paper AW5L.5.
- [2] Ferreira de Lima, Thomas, Doris, Eli A., Bilodeau, Simon, Zhang, Weipeng, Jha, Aashu, Peng, Hsuan-Tung, Blow, Eric C., Huang, Chaoran, Tait, Alexander N., Shastri, Bhavin J. and Prucnal, Paul R.. "Design automation of photonic resonator weights" Nanophotonics, vol. 11, no. 17, 2022, pp. 3805-3822.

Version 1:

Reviewer comments:

Reviewer #1

(Remarks to the Author)

Authors have addressed all my comments properly in revised manuscript.

Reviewer #2

(Remarks to the Author)

Thank you for addressing the comments.

Reviewer #3

(Remarks to the Author)

The authors have addressed my comments in the modified manuscript. No further comment.

Response to CommsEng Review Comments

May 2025

Response to Reviewers

We thank all reviewers for their careful reading of our manuscript and their constructive comments. We have addressed all comments as detailed below. Significant changes in the revised manuscript are highlighted in blue font.

Reviewer 1

Comment 1.1: I suggest avoid saying “... present a new approach to optical beamforming using microring weight banks as RF phase shifters ...”. The MRRs are not actual RF phase-shifters and they only adjust the weights. So please make sure similar sentences are edited in next revision.

Response 1.1: We believe the wording is correct as the MRRs are indeed used to apply a phase shift to the RF signal by independently weighting the signal quadratures. In the context of RF electronics it is common to refer to these components as phase shifters. Please see reference [17] of our manuscript.

Comment 1.2: Details on photonic chip and devices need to be more complete. For instance, radius and Q-factor of MRRs, FSR, heater efficiency, Bandwidth of photodiodes, grating couplers loss, etc.

Response 1.2: As suggested, we have added all of the requested details in the new Methods Section 5.3 with details of the chip and testing

Comment 1.3: Authors should discuss the choice and trade-offs of MRR’s Q-factors on the system performance. For instance, the Q-factor should be low-enough to accommodate the RF signal bandwidth, at the same time it impacts the MRR weights loss. Please include a section on design trade-offs of photonic devices specially MRRs.

Response 1.3: Following your suggestion, we have updated the section “Challenges and Design Considerations” to include a discussion of MRR design tradeoffs. Hugh and Bhavin, please review.

Comment 1.4: In terms of overall practicality of proposed system (Fig. 3), one needs to consider all pros/cons vs. just distributing the signals optically and using electrical RF phase-shifters. For instance, total signal loss needs to be estimated and compares between these methods. (Notice that IQ signal amplitude is at 0.4 max!). Such an analysis needs to be included.

Response 1.4: We recognize that a balanced discussion of both advantages and disadvantages is essential. Accordingly, Section 2.1 now includes a comparison with a repartitioned system leveraging WDM signal distribution with RF electrical phase shifting.

Comment 1.5: There's a discussion on SiN vs Si, and even use of amplifiers. This is misleading because no clear estimate of required optical/RF power is mentioned. Include some rough estimates for these, so that the readers can know the limitations/requirements of proposed approach.

Response 1.5: We understand the concern. Since the optical power requirement is dependent on the application and use case, we have included some rough estimates based on the optical power requirements at the detector in Section 3.1.

Comment 1.6: Notice that while MRRs are thermally sensitive, it is essential (due to process variations) to have some tuning capability with large-range to initially adjust MRRs to laser lines. Make sure this is clear in your discussion. The discussion on MRR's FWHM is incomplete because RF signal BW is not discussed. Also, number of wavelengths depends on FSR as well. Please edit these sections.

Response 1.6: Following your comment, we have included a note on the tuning range requirement in the discussion, and updated section "Challenges and design considerations" to include a discussion of the RF signal bandwidth in relation to the FWHM of the rings.

Comment 1.7: Which foundry has been used? There must be a chip fabrication and testing in Section 5 of Methods.

Response 1.7: We have added the details of the AMF foundry in a new Section 5.3 called "Chip Fabrication and Testing".

Reviewer 2

Comment 2.1: Abstract: Phase error of better than 2° — the degree symbol looks weird.

Response 2.1: We corrected the formatting.

Comment 2.2: Figure 2.1 should be Figure 3?

Response 2.2: Thank you pointing out this error. The figure labels have been corrected.

Comment 2.3: Feedforward control is interesting but needs more elaboration. What exactly is the rule and what can be extended?

Response 2.3: As suggested, we have elaborated on the calibration method of the phase shifter in Section 2.3.

Comment 2.4: Fig 6 (a) and (b) are reversed with respect to the text description in section 2.4.

Response 2.4: Thank you for noting this. We fixed the error.

Comment 2.5: Discussion: It's okay to compare the compact form factor of MRRs with other optical implementations using optical delay lines or dispersive fibers. However to compare with electrical solution, the entire optical system including the MZMs and the multiplexers should be

included, both are quite large. Can the authors elaborate on this comparison and the scalability of the entire system, not only the MRR weight bank? Can comb-driven DWDM be used? Doesn't have to be multiple laser sources with a multiplexer.

Response 2.5: In response to your comment, we have revised the discussion in Section 3.2 to include a coarse estimate of the total system footprint, accounting for the modulators, multiplexers, rings and lasers.

Comment 2.6: Scalability analyses: what limits the maximum N_t and N achievable? The analysis seems quite qualitative. At least some quantitative analysis using analytical models is desirable. Other factors limiting scalability could include power and control of the laser array, multiplexer and splitter design, MRR free-spectral range, accumulated loss on bus, etc, if the author could comment on as well.

Response 2.6: Per your suggestion, we have revised Section 3.1 of the discussion to comment on the primary factors that limit the scalability of the MRR weight banks in this application.

Reviewer 3

Comment 3.1: Authors should discuss the significant advances that they have presented in the current manuscript as compared to Ref 1. Figure 10 of the current manuscript seems identical to Figure 3 of Ref.

Response 3.1: The previous work was a brief conference paper introducing the technique of MRR-based RF phase shifting at a conceptual level. In this manuscript we proposed a hybrid beamforming system leveraging the MRR-based phase shifter introduced in the previous work. We present experimental results of the proposed beamformer, and discuss further improvements to address limitations of MRRs for RF signal processing. The contributions of this work are highlighted at the end of Section 1.

Comment 3.2: The concept and physics of 'micro-ring resonator (MRR) weight banks' should be briefly introduced in the manuscript along with their advantages compared to other state-of-the-art photonic systems [2].

Response 3.2: As suggested, we have updated the Introduction to include a brief discussion of the function and advantages of MRR weight banks.

Comment 3.3: The operating principle of the proposed technique should be discussed as a subsection for broad visibility and understanding of the technique to the readers.

Response 3.3: We discuss the system architecture of the proposed technique in Section 2.1 of the original manuscript, which includes a description of the signal transformation through the system. We have also updated the Introduction to include a description of the operating principle of MRR weight banks.

Comment 3.4: Authors should discuss the broad implications of their approach for photonic chip design and photonic signal processing; and how one can potentially incorporate artificial intelligence (AI) into the architecture to realize a more versatile and automated system. Given the authors discussed wireless beamforming, can authors comment on the polarization sensitivity of their approach? With a photonics-based approach, can authors realize polarization diversity

and/ or structured (polarization and phase-engineered) beamforming? A detailed discussion on this aspect is required. Authors should also comment on whether their architecture can provide beamforming for above 100 GHz+ frequencies.

Response 3.4: We appreciate the suggestion to address broader implications; however, we consider the mentioned topics to be beyond the scope of this work. We did not consider synergies with artificial intelligence as our research focused on augmenting conventional RF signal processing systems with photonic technologies.

Our architecture concerns the weight-and-sum mechanism of the beamformer and does not affect the polarization sensitivity of the system. RF beamformers achieve polarization insensitivity using dual polarized antenna elements with separate signal paths for each polarization. The implementation of the phase shifters and combiners would have no bearing on the polarization sensitivity of the system.

We did not consider the scalability of our architecture at ultra-high frequencies. This would be challenging due to the high bandwidth requirements that would be imposed on the MRRs. The modulation scheme may need to be reconsidered to extend this technique beyond the mmWave bands. Following your comment, we have made the frequency range that we consider directly suitable explicit in the first paragraph of Section 3 in the revised manuscript, and we updated the discussion in Section 3.1 to comment on the challenges associated with high frequency operation.

References

- [1] M. Nichols, M. Salmani, E. Luan, A. Eshaghi, and L. Lampe, "Photonic Vector Modulator Based on MRR Weight Banks," in Conference on Lasers and Electro-Optics, Technical Digest Series (Optica Publishing Group, 2022), paper AW5L.5.
- [2] Ferreira de Lima, Thomas, Doris, Eli A., Bilodeau, Simon, Zhang, Weipeng, Jha, Aashu, Peng, Hsuan-Tung, Blow, Eric C., Huang, Chaoran, Tait, Alexander N., Shastri, Bhavin J. and Prucnal, Paul R.. "Design automation of photonic resonator weights" *Nanophotonics*, vol. 11, no. 17, 2022, pp. 3805-3822.